# Unexpected large impact of small charges on surface frictions with similar wetting properties

Chunlei Wang[1,2,6], Haijun Yang[1,2,6], Xian Wang[3], Chonghai Qi[1,4], Mengyang Qu[1], Nan Sheng[1,2], Rongzheng Wan[1,2], Yusong Tu[3✉] & Guosheng Shi[5✉]

Generally, the interface friction on solid surfaces is regarded as consistent with wetting behaviors, characterized by the contact angles. Here using molecular dynamics simulations, we find that even a small charge difference ($\leq 0.36$ e) causes a change in the friction coefficient of over an order of magnitude on two-dimensional material and lipid surfaces, despite similar contact angles. This large difference is confirmed by experimentally measuring interfacial friction of graphite and $MoS_2$ contacting on water, using atomic force microscopy. The large variation in the friction coefficient is attributed to the different fluctuations of localized potential energy under inhomogeneous charge distribution. Our results help to understand the dynamics of two-dimensional materials and biomolecules, generally formed by atoms with small charge, including nanomaterials, such as nitrogen-doped graphene, hydrogen-terminated graphene, or $MoS_2$, and molecular transport through cell membranes.

[1] Shanghai Institute of Applied Physics, Chinese Academy of Sciences, Shanghai 201800, China. [2] Shanghai Advanced Research Institute, Chinese Academy of Sciences, Shanghai 201210, China. [3] College of Physics Science and Technology, Yangzhou University, Jiangsu 225009, China. [4] School of Physics, Shandong University, Jinan 250100, China. [5] Shanghai Applied Radiation Institute and State Key Lab. Advanced Special Steel, Shanghai University, Shanghai 200444, China. [6] These authors contributed equally: Chunlei Wang, Haijun Yang. ✉email: ystu@yzu.edu.cn; gsshi@shu.edu.cn

The microscopic nature of solid/liquid boundary friction has long been a subject of interest in materials and biology systems. It relates to the lubrication and nano-tribology of materials[1–10], developing micro-fluidic and nano-fluidic devices[11–19], the motion of biological molecules[20–22] and even protein folding[23]. Generally, the microscopic friction is usually determined by measuring the contact angle[24–34]; a large contact angle indicating a hydrophobic surface is associated with low surface friction, and vice versa[35]. However, the contact angle as a macroscopic surface wetting property is not always consistent with the microscopic viewpoint, even for nano-scale wetting behaviors themselves on polar surfaces[32–34,36]. This complicates the relationship between surface friction and surface hydrophobicity/hydrophilicity.

Charges below 0.36 e widely exist in atoms or groups of popular two-dimensional materials and biomolecules. For example, the carbon atoms in N-doped[37,38] and the hydrogen-terminated graphene[39] usually attain the charge from 0.10 e to 0.20 e, the carbon atoms of the terminal methyl of a lipid usually attain the charge of −0.18 e[40–42], the carbon atoms of benzene rings in Phe group of protein residues can attain a charge of −0.115 e[43,44], and the S atoms of $MoS_2$ have a charge of −0.36 e[45]. Such small charges on the solid surfaces are usually neglected in studying the surface dynamics. This is partly because total surface–water interactions due to small charges have negligible electrostatic interaction energies, which is confirmed by the similar contact angles on these surfaces. Unfortunately, the association of solid–water interfacial friction and the wetting properties of solid charged surfaces still remain unknown.

In this work, based on the molecular dynamics simulations, we unexpectedly reveal a change in the friction coefficient of over an order of magnitude within a small charge range (≤0.36 e) even for similar contact angles. This is further confirmed by experimentally measuring interfacial friction of graphite and $MoS_2$ contacting on the water using atomic force microscopy (AFM). The large variation in friction coefficient is attributed to the different fluctuations of localized potential along the solid surface due to the charge distributions. Meanwhile, the surface–water interaction energy per unit area almost remains constant. Particularly, we find a linear relationship between the surface friction coefficient and the fluctuations in the localized potential energy profile, $(\Delta E_{micro})^2$ due to the surface charge. These results help clarifying the physical mechanism relating macro wetting behaviors and micro frictions of various surfaces, including biological surfaces.

## Results and Discussion

### Contact angle and frictions on a model charged solid surface.
Figure 1a shows the geometry of the solid surface system with a small charge in the range 0 e–0.36 e. Positive and negative charges of the same magnitude q were assigned to atoms located diagonally on neighboring hexagons. A square-lattice model surface with a similar charge range can be found in Supplementary Note 4 of the Supplementary Information. A square-lattice model surface with a similar charge range can be found in Supplementary Note 4 in Supplementary Information. Overall, the modeled solid surface was neutral. The contact angle was determined by fitting the curve of the liquid–vapor interface, according to the method in previous work[32,46]. More details on the simulation setup can be found in the Methods section. The friction coefficient was calculated using the Green–Kubo relationship[6,35,47]:

$$\lambda = \frac{1}{Ak_B T} \int_0^\infty dt \langle F(t)F(0) \rangle_{equ}, \quad (1)$$

where $F(t)$ is the total tangential force acting along the x direction on a surface with area A and the average runs over equilibrium configurations; $k_B$ is the Boltzmann constant, T is the temperature, and λ is the friction coefficient for water on various solid surfaces for sufficiently long time intervals (1 ps), as shown in Supplementary Fig. 1 and described in Supplementary Note 1.

Figure 1b (black columns) shows the relative friction $\lambda_q/\lambda_0$ as a function of the charge within the small range 0 e ≤ q ≤ 0.36 e, where $\lambda_q$ is the friction coefficient at a particular charge q and $\lambda_0$ is the friction on a neutral surface calculated with the Green–Kubo relationship. The calculated friction increases from $1.2 \times 10^4$ Ns/m³ for q = 0 to $4.9 \times 10^5$ Ns/m³ for q = 0.36 e; the latter friction is over one order of magnitude (41 times) higher than the former. Note that this friction for q = 0.36 e is quite close to the value of ~$2.0 \times 10^5$ Ns/m³ on $MoS_2$[45] where the surface charge of the S atoms is −0.36 e. Figure 1b (right wine columns) shows the dependence on contact angle of the surface charge of the hexagonal lattice. When q ≤ 0.36 e, the contact angle slightly decreases from θ = 88° ± 2° for q = 0 e to θ = 68° ± 2° for q = 0.36 e, which is consistent with the fact that surface atomic charges slightly affect wetting within a small charge range[48]. Huang et al.[35] proposed that the friction coefficient $\lambda^*$ closely related to the contact angle θ according to

$$\lambda^* \propto (1 + \cos\theta)^2. \quad (2)$$

Figure 1b (red textured columns) shows that the relative friction $\lambda_q^*/\lambda_0^*$ based on the contact angles varies by only 50% as q increases from 0 e to 0.36 e. This clearly shows that a small surface charge (q ≤ 0.36 e) significantly affects the friction coefficient even similar contact angles, in contrast to the conventional viewpoint. This deviation of very high calculated friction from the low theoretical friction based on contact angles is universal, even on a square-lattice polar surface with a very small charges (see Supplementary Note 4 in the Supplementary Information).

To further understand the origin of the large change in friction coefficient, we studied the microscopic water-surface interaction in the first contact layer with the thickness of 0.50 nm (see more details of the water density in Supplementary Note 2 and Supplementary Fig. 2 in the Supplementary Information). We divided the area into small squares each with an area of 0.25 Å² and calculated the average microscopic interaction energy $E_{micro}$ of each water molecule and of all the solid surface atoms. Figure 2a shows a continuously smooth $E_{micro}$ distribution over the area at q = 0 e, with almost no energy barrier to obstruct the motion of the water molecules. When q increases to 0.2 e, the colored area becomes discontinuous and separated by high and low energies. This means that the water molecules encounter energy barriers when they move, resulting in higher friction. We have plotted the distribution of $E_{micro}$ in Supplementary Fig. 3 of the Supplementary Information, where $E_{micro}$ has a broader distribution when the charge increases. We calculated the average total solid–water interaction energy E and found that it slightly decreases from −56 kJ mol⁻¹nm⁻² to −68 kJ mol⁻¹nm⁻² as q increases from 0 e to 0.36 e. This is consistent with the variation of the contact angle[29] but quantitatively inconsistent with the very high change in friction coefficient (a factor of 41) over this range of q.

The microscopic potential energy profiles of square-lattice surfaces for different charge value q = 0 e (a), 0.1 e (b), 0.2 e (c) and 0.3 e (d) are shown in Supplementary Fig. 5. The dependence of the microscopic energy variance $(\Delta E_{micro})^2$, the surface friction

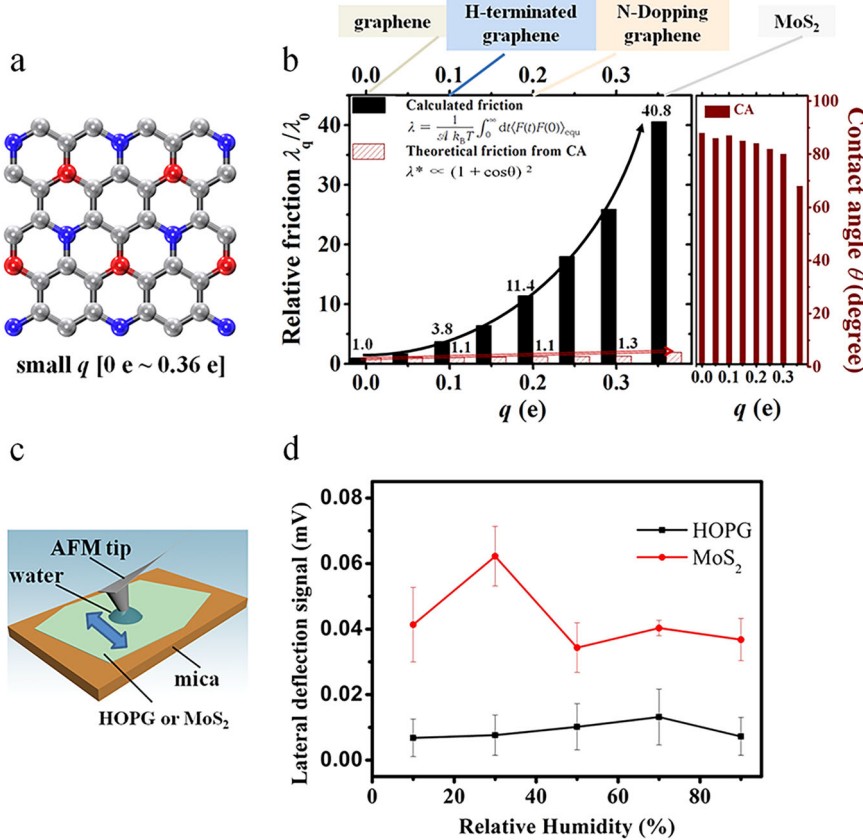

**Fig. 1 Friction coefficients for different surfaces.** A hexagonal solid surface, the highly oriented pyrolytic graphite (HOPG) and $MoS_2$ solid surfaces are shown. **a** Model hexagonal solid surface with small charge ranging from 0 to 0.36 e. The red and blue spheres represent positive and negative charges, respectively, and the gray spheres represent neutral atoms. **b** Estimated relative friction coefficient $\lambda_q/\lambda_0$ ($q \leq 0.36$ e) of liquid water on the solid surface calculated with the Green–Kubo relationship (black solid columns), and relative friction calculated from the contact angle using the theory proposed by Huang et al. in ref. [35] (red textured columns), together with the contact angles in respect of the charge $q$ values from 0.00 e to 0.36 e. When $q$ increases from 0.00 e to 0.36 e, the Green–Kubo relative friction coefficient changes by a factor of 41 while the coefficient calculated from the contact angle changes by 50%. This panel also shows the typical materials with the atomic charge range from 0.00 e to 0.36 e, including graphene, hydrogen-terminated graphene, N-doped graphene, and $MoS_2$. **c** Schematic for the measuring friction with an AFM tip. The arrow shows the moving direction of the AFM tip, which is perpendicular to the cantilever. A water droplet condenses from the humid environment between the hydrophilic AFM tip and the HOPG or $MoS_2$. **d** Lateral deflection signal of the AFM cantilever when rubbing on HOPG or $MoS_2$, respectively, varying with the relative environmental humidity. Error bars are the standard deviations of the lateral deflection signals with the "discrete frequency" larger than 200. The friction force is proportional to the lateral deflection signal".

coefficient and the analysis of total interactions between surface-water on the surface charge $q$ is discussed in Supplementary Note 5.

The large variation in friction coefficient is due to the large fluctuation of the localized potential energy, defined as $(\Delta E_{micro})^2 = \langle (E_{micro} - \langle E_{micro} \rangle)^2 \rangle$, which is obtained from the standard deviation of the microscopic interaction energy $E_{micro}$ under inhomogeneous charge distributions. We have established a quantitative relationship between the friction coefficient and $(\Delta E_{micro})^2$, which is obtained from the standard deviation of the microscopic interaction energy $E_{micro}$. We have found that the surface friction increases monotonous as the $\Delta E_{micro}$ increases. Figure 3 shows a good linear relationship between the simulated friction coefficient and $(\Delta E_{micro})^2$ obtained with a fitting factor $\zeta \sim 130$ when $\Delta E_{micro}$ is less than 0.45 kJ/mol. On the basis of the Green–Kubo relationship of Eq. (1)[35], the surface friction $\lambda$ is proportional to the square of the total surface-water tangential force, $\langle F \rangle^2$. The charge $q$ of each polar solid surface uniquely determines $\langle F \rangle^2$. This is different from previous work in which $\langle F \rangle^2$ is proportional to the Lennard-Jones energy reflecting the solid–water interactions on atomic surfaces[35]. The force $\langle F \rangle$ is

roughly proportional to the potential energy fluctuation $\Delta E_{micro}$; thus $\lambda \sim (\Delta E_{micro})^2 \sim q^2$ (see the further discussion in PS 5 of the Supplementary Information). Thus, the local potential fluctuation and friction coefficient quadratically increase as $q$ increases (see also Supplementary Figs. 6 and 7 in the Supplementary Information), which also explains the increase in the friction coefficient of $MoS_2$ compared with that of graphene[6,45]. The linear relationship between $\lambda$ and $(\Delta E_{micro})^2$ is universal. For a square-lattice surface with a charge of around 0.3 e, we have found that the friction coefficient changes by a factor of 1000 even with a similar contact angle (see Supplementary Note 3 and Supplementary Fig. 4 in the Supplementary Information). Despite different surface lattices, Fig. 3 (black solid squares) shows a linear relationship between $(\Delta E_{micro})^2$ and $\lambda$ with almost the same factor of 130.

**Contact angle and frictions on a bimolecular solid surface.** In addition, we experimentally measured the surface friction of highly oriented pyrolytic graphite (HOPG) and $MoS_2$ contacting on the water by rubbing a hydrophilic AFM tip with a

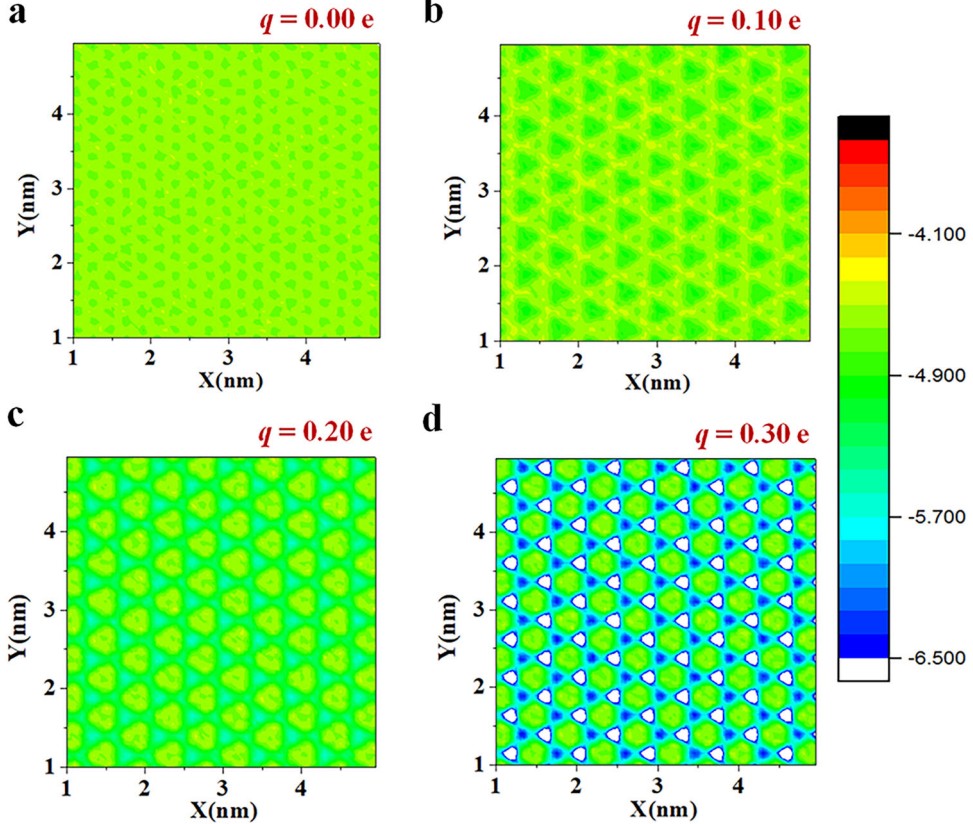

**Fig. 2 Microscopic potential energy profiles of solid surfaces.** The various charges of 0 e (**a**), 0.1 e (**b**), 0.2 e (**c**), and 0.3 e (**d**) for hexagonal lattice solid surfaces are presented, respectively. The potential energy profile is calculated for each water molecule in the first layer within an area of 0.25 Å$^2$.

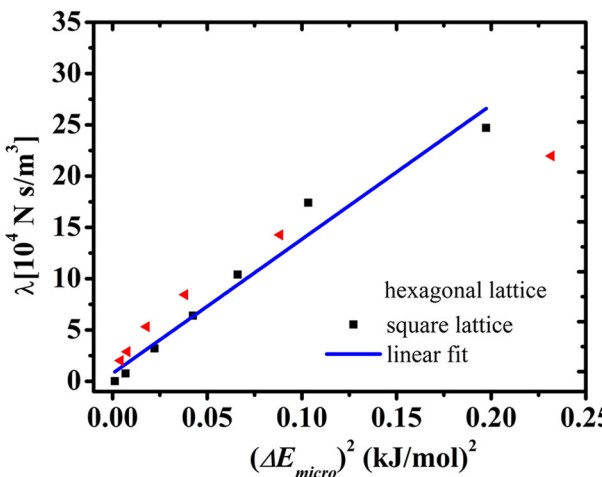

**Fig. 3 Friction coefficient versus fluctuation of the localized potential energy.** Linear relationship between the surface friction coefficient and the fluctuation in microscopic potential energy, $(\Delta E_{micro})^2$, for the hexagonal (red) and square (black) lattice surfaces for various charges. The linear fit is shown in blue.

constant loading force (see Fig. 1c and more details in the Methods). The average the lateral deflection signal on $MoS_2$ fluctuates around 0.04 mV at different relative environmental humidity, about 6 times of that on HOPG, as shown in Fig. 1d. It indicates that the surface friction of $MoS_2$ is about 6 times of

that of HOPG since the friction force is proportional to the lateral deflection signal. We also measured the contact angles to be $83° \pm 5°$ and $83° \pm 6°$ on HOPG and $MoS_2$, respectively. These experimental results support our simulation of the large variation of solid–water interfacial friction coefficient on the two two-dimensional material surfaces with the small charge and similar contact angles. We note that the experimental difference in friction between $MoS_2$ and HOPG is less than in the simulation, which may be partly due to the different thicknesses of the water films and different lattices of these two substrate surfaces.

We note that many beta-carbon atoms in biological molecules have charges of less than 0.36 e, such as $-0.115$ e for the carbon atoms of benzene rings in Phe, $-0.18$ e for the methyl carbon atoms in Ala, $-0.22$ e for the sulfur atoms in Cys, and $-0.32$ e for the carbon atoms in benzyl methyl phosphonate[49]. We expect that these low charges do not affect the hydrophobicity of the backbone but greatly affect the friction of the backbone-water interface. To verify this, we simulated a surface composed of rigid methylene $CH_2$ groups (see Fig. 4a) with $q_C = -0.12$ e and $q_H = 0.06$ e, which is an important backbone in lipid molecules. By rescaling the charges from 0.06 e to 0.36 e (with $q_C = -0.12$ e corresponding to a methyl carbon atom in a lipid), we find that the variation in surface friction coefficient can even reach a factor of 4 for surfaces with similar contact angles (Fig. 4b). However, Eq. (2) yields a variation in relative friction $\lambda^*$ of only 10%, in sharp contrast to the factor of 4 for solid surfaces as $q_C$ increases from 0.06 e to 0.36 e. Again, the small charge may not affect the hydrophobicity of a biological backbone composed of $CH_2$

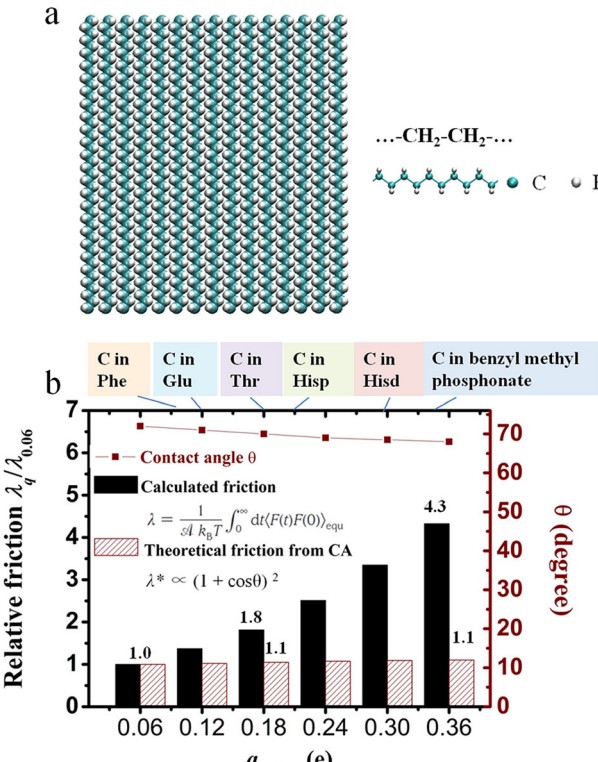

**Fig. 4 Relative friction coefficient for biological surface with CH₂ groups.**
**a** Plane model of a carbon-based solid surface composed of CH₂ groups.
**b** Estimated relative friction coefficient $\lambda_k/\lambda_{0.06}$ of liquid water on the solid surface simulated with the Green–Kubo relationship (black solid columns), relative friction calculated from contact angles in ref. [35] (red textured columns), and contact angles (wine squares) for various atoms with charges of less than 0.36 e on solid surface composed of CH₂ groups. This panel also shows the various small charge carbon atoms less than 0.36 e in some typical biological protein residues.

groups, but greatly affects the surface friction, which may be the key to protein folding and transport of biological molecules.

In summary, we have found a change in friction coefficient of over an order of magnitude over a small charge range (≤0.36 e) on two-dimensional materials and lipid surfaces even with little change in contact angle. We have confirmed this by experimentally measuring the solid/water surface friction of graphite and MoS₂ using AFM. This huge difference in interfacial friction is caused by the different fluctuations in localized solid–water interaction potentials along the solid surface due to the heterogeneous charge distributions. Different structures such as hexagonal and square lattices and lipid alkane groups exhibit similar deviations in contact angle and surface friction, suggesting that the phenomenon is general. More importantly, atoms with small charge are very common in materials such as nitrogen-doped graphene and MoS₂. This work will thus help in accurately understanding the dynamics of these materials and facilitate their fabrication or manipulation.

Our findings show that general wetting behavior cannot be directly used to infer the friction of biological molecules. This requires generating detailed energy profiles along the biological surface contacting with an aqueous substance to determine the microscopic behavior at the solid–liquid interface. Considering that the charges of most atoms in biological molecules are lower than 0.36 e (67.1% of all residue atoms listed in the OPLS-AA force fields[49]), our findings take a major step in understanding

protein–ligand binding[50–52] in crowded cell environments, molecular transport through lipid membranes[40], and protein folding related to internal friction.

## Methods

**Molecular dynamics simulation details and parameters.** The periodic boundary conditions were applied in all directions. Our MD simulations were performed using a time step of 1.0 fs in an NVT ensemble at a temperature of 300 K. The solid atom with Lennard-Jones parameters $\varepsilon_{ss} = 0.2325$ kJ/mol, $\sigma_{ss} = 3.4$ Å[53], and the SPC/E water model[54] were used. A cutoff of 1.0 nm was used to calculate the dispersion (van der Waals) energies. The particle-mesh Ewald (PME)[55] method and a real-space cutoff of 10 Å were used to treat long-range electrostatic interactions. The size of the simulation box is $x = 6.395$ nm, $y = 6.816$ nm and $z = 21$ nm. The integration time step in a simulation was 1 fs. All MD simulations were carried out using the Gromacs 4.5[56] software Package. We performed two series of simulations to study the contact angles and the surface frictions coefficients. The simulations to obtain the contact angles consists 900 water molecules while the simulations to obtain surface frictions coefficients consists 5103 water molecules with the thickness of water film as 4 nm. The simulation time was 80 ns for each system, and the last 2 ns data were collected for analysis.

**Experimental measurement of the friction force and contact angle on HOPG or MoS₂.** In an environmental chamber (Shanghai Espec Environmental Equipment Corporation), a Multimode-8 AFM with Nanoscope V controller and J scanner (Bruker, Santa Barbara, CA, USA) was used in contact mode for all the measurement of the friction force. The nominal spring constant, thickness, tip height, cantilever length, and cantilever width of AFM cantilevers (XSC11 Al/BS, MikroMasch, Estonia) are 42 N/m, 2.7 μm, 15 μm, 100 μm, and 50 μm, respectively. For engaging the AFM tip, the "Deflection setpoint" was set 0.5 V, and the "VERT", "HORZ", and "SUM" on the AFM base kept $0.50 \pm 0.01$, $0.00 \pm 0.02$, and 3.73, respectively. After tip engaging, the "Deflection setpoint" was increased to 0.7 V to increase the loading force on the substrate through the AFM tip. Through this kind of operation process, we believe the loading force is fixed for all the experiments. By rubbing the AFM tip on the substrate at a scan angle of 90° (i.e. orthogonal to the long axis of the cantilever, Fig. 1c)[57], the height image and friction signal images in both the trace and retrace directions were collected simultaneously at a controlled relative humidity ranging from 10–90%.

After subtracting the retrace frictional image from the simultaneously collected trace image, an image of the friction hysteresis at a given sample area can be obtained, which is considered to be twice of the friction force exerted at that point[58] and can be exported as an ASCII file by its built-in software for statistics. To exclude the noise in the experiments, only the friction forces with the "discrete frequency" larger than 200 were adopted to calculate their averages and standard deviations. The friction force ($V_{lateral}$) takes mV as unit.

All contact angles were measured by an Attension Theta optical goniometer (Biolin Scientific AB, Sweden). A HOPG or MoS₂ sheet freshly cleaved with a Scotch tape was adopted as the substrate and placed onto the level sample stage between the light source and the camera of the goniometer. A drop (about 4 μl) of pure water (18.2 MΩ cm) was injected onto the substrate. A side profile photograph of the sessile droplet was captured by the goniometer and analyzed with its built-in software to determine their static contact angles. The experiment for each substrate was repeated more than 10 times, and the contact angles at the left and right side of the droplet were averaged together.

## Data availability

The data supporting the findings of this study are available within the article and its Supplementary Information files, or from the corresponding authors on reasonable request.

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

## Acknowledgements

The authors thank Prof. Haiping Fang for his inspiring idea and discussions. We also thank David Deibert for improving the English of this paper. This study was supported by the National Natural Science Foundation of China (Nos. 11674345, 11675138, U1532260, U1632135, U1932123), the National Science Fund for Outstanding Young Scholars (No. 11722548), Key Research Program of Chinese Academy of Sciences (QYZDJ-SSW-SLH019), Innovative research team of high-level local universities in Shanghai, Shanghai Supercomputer Center of China, Computer Network Information Center of Chinese Academy of Sciences, and Special Program for Applied Research on Super Computation of the NSFC-Guangdong Joint Fund (the second phase).

## Author contributions

G.S., Y.T., and C.W. designed the project. C.W., X.W., C.Q., M.Q. performed theoretical computation; H.Y., G. S., and C.W. performed the experiment; C.W., G.S., Y.T., and H.Y. analyzed the data; X.W., C.Q., M.Q., N.S., and R.W. conducted some theoretical analysis; C.W., G.S., Y.T., and H.Y. co-wrote the paper. All authors discussed the results and commented on the manuscript.

## Competing interests

The authors declare no competing interests.
