## [Peer Review File · Communications Chemistry]

Reviewers' comments:

Reviewer #1 (Remarks to the Author):

The manuscript was focused on "Large Impact of Small Charges on Surface Frictions". The manuscript was prepared very well and the work is important but before publish, some points should be considered:

- 1) In this work a Lennard-Jones potential was considered for substrate charged atoms with hexagonal structure. I think this simple interatomic potential cannot maintain the geometrical stability of hexagonal structure with charged atoms. I expect the respected authors to present evidence about stability of the substrate structure. Furthermore which kind of interatomic potentials was considered between water molecules and substrate atoms?
- 2) What is the size of simulation box? This subject has to be declared in the manuscript. Because in Z direction the length of simulation box have to be enough large to avoid the destructive effects of periodic boundary condition in Z direction.
- 3) In lines 126 -157, the fluctuations of ϵ_{micro} is considered as a criteria for friction. In other word, high fluctuation is equal to high friction. The authors have to show that the area division cannot affect the result. I mean that is there any reason for selection of 0.25 \AA^2 for each square?
- 4) What long-range contribution corrections were employed for dispersion and electrostatics "The authors now state that they used Ewald summation for long-range electrostatics; they do not comment on long-range dispersion. This should be clarified (i.e., stated in the manuscript). I have no objection to neglecting long-range dispersion in this case, but please say so if you do it, so that readers of the paper can understand what it is that you did here

Reviewer #2 (Remarks to the Author):

This manuscript reveals an interesting fact that the surface frictions has little relationship with the surface wetting behaviors in the micro scale. The authors used molecular dynamics simulation to figure out the linear relationship of surface friction with square of energy fluctuation, which is also confirmed by the experimental atomic force microscopy method. I think this manuscript is suitable for publication in Communications Chemistry before the below concerns is addressed.

1. In line 43, the authors demonstrate the value less than $0.35e$, but in line 49, they take an example of $0.36e$. The authors should clarify this.
2. The method of calculating ϵ_{micro} should be included in the manuscript, at least introduced in supporting information.
3. In the work of Xu et al, they constructed similar surface model, but test the nanofluidic behaviors. In that work, the surface charge ranges from 0.0 to $3.0e$, but their surface friction, which is calculated from non-equilibrium MD simulations, does not change as much as this work demonstrates. The authors might give some discussions on this. (Journal of Physical Chemistry C, 2018, 122: 15772-15779.)
4. The measurement of contact angle in MD simulations are still controversial. Since the wetting behavior is somehow determined by the pressure (Research, 2019, 2019: 10.). Did authors consider the influence of pressure on the contact angle values? Or please convince me that the pressure takes no effect in this work.
5. I am interested in the change of surface frictions if the surface atom density is increased or decreased. Can authors give some discussions on this? The simulation data are not required to provide.

6. The acronym of HOPG should be introduced in the text of manuscript although it appears in Figure 1.

Reviewer #3 (Remarks to the Author):

This is an interesting paper discussing the effect of surface charging on friction. The obtained results are useful, but the information is not surprising and particularly innovative. On the other hand, the obtained correlations between various system parameters are carefully examined, which makes this paper a nice reference work. Overall, I feel that the paper would fit better to Scientific Reports.

Reviewer #1 (Remarks to the Author):

The manuscript was focused on “ Large Impact of Small Charges on Surface Frictions”.

The manuscript was prepared very well and the work is important but before publish, some points should be considered:

1) In this work a Lennard-Jones potential was considered for substrate charged atoms with hexagonal structure. I think this simple interatomic potential cannot maintain the geometrical stability of hexagonal structure with charged atoms. I expect the respected authors to present evidence about stability of the substrate structure. Furthermore which kind of interatomic potentials was considered between water molecules and substrate atoms?

Response: Thanks for the comments. We agree that only the Lennard-Jones potential cannot maintain the geometry and the stability of this kind of two-dimensional materials. Usually, the similar two dimensional materials as our two-dimensional model surface, such as graphene, boron-nitrogen (BN) and the nitrogen-boron-dopping graphene, extensively exist. For these materials, the two-body (covalent bond-stretching,), three body (angle-bending,) and four body potentials (improper dihedrals) are chosen to maintain the geometry of the solid surfaces. In this work, we also used these potential parameters for our model surfaces just as the graphene. Simultaneously, the atoms of the solid surface are fixed by freezing the atoms in our model solid surface. This method was also adopted in the previous works [J. Phys. Chem. C 115, 3018 (2011); J. Phys. Chem. C 2012, 116, 15962 (2012)]. This setup can help us capture the the effect of small charges on the surface wetting and the surface frictions, and also can keep the stability of the substrate structure.

As for the questions “*interatomic potentials was considered between water molecules and substrate atoms*” by referee, the interatomic potentials were the Lenard-Jones (LJ) and electrostatic potential. The LJ potential is:

$$V_{LJ}(r_{ij}) = 4\epsilon_{ij} \left(\left(\frac{\sigma_{ij}}{r_{ij}} \right)^{12} - \left(\frac{\sigma_{ij}}{r_{ij}} \right)^6 \right) \quad (1),$$

where the energy parameter of the solid atoms is $\epsilon_{ss} = 0.2325$ kJ/mol and the $\sigma_{ss} = 0.34$ nm. According to the combination rule, the interaction parameter between water and the solid surface atoms is $\epsilon_{ws} = (\epsilon_{ss} \cdot \epsilon_{ww})^{1/2} = 0.6502$ kJ/mol and the $\sigma_{ws} = (\sigma_{ss} \cdot \sigma_{ww})^{1/2} = 0.3166$ nm. The electrostatic potential is

$$V_c(r_{ij}) = \frac{1}{4\pi\epsilon_0} \cdot \frac{q_i q_j}{\epsilon_r r_{ij}} \quad (2),$$

where the charge on the solid surface is from 0 e to 0.36 e, while the water molecules we used is SPC/E water model with $q_O = -0.8476$ e and $q_H = 0.4238$ e [H. J. C. Berendsen, J. R. Grigera, and T. P. Straatsma, J. Phys. Chem. 91, 6269 (1987)].

2) What is the size of simulation box? This subject has to be declared in the manuscript. Because in Z direction the length of simulation box have to be enough large to avoid the destructive effects of periodic boundary condition in Z direction.

Response: Thanks for the suggestions. The simulation box was set as $x = 6.395$ nm, $y = 6.816$ nm and $z = 21.000$ nm. We agree with the comment of the referee, the z axis distance is set as three times larger than both the x and y simulation box size to avoid the imagine effect.

Change made: We have added this information in “PS 6. Simulation details and parameters we adopt” of our Supporting Information, which reads “*The size of the simulation box is $x = 6.395$ nm, $y = 6.816$ nm and $z = 21$ nm.*” in line 8 of the PS 6 of the Supporting Information.

3) In lines 126 -157, the fluctuations of Emicro is considered as a criteria for friction. In other word, high fluctuation is equal to high friction. The authors have to show that the area division cannot affect the result. I mean that is there any reason for selection of 0.25 \AA^2 for each square?

Response: Thanks for the comments. In this manuscript, we chose the 0.25 \AA^2 for

each square with the width of the square of 0.5 \AA , which is less than the minimum distance of neighboring atoms (0.142 nm in this manuscript). We have tested the other choice of the area to calculate the fluctuations of E_{micro} , for example, $S = 0.09 \text{ \AA}^2$, 0.16 \AA^2 , 0.36 \AA^2 , 0.49 \AA^2 . The results are listed in Table R1. We have found that the selection of the area does affect the fluctuation of the localized potential energy ΔE_{micro} , which are typically 0.026 kJ/mol for $q = 0 \text{ e}$, 0.020 kJ/mol for $q = 0.1 \text{ e}$, 0.025 kJ/mol for $q = 0.2 \text{ e}$ and 0.023 kJ/mol for $q = 0.3 \text{ e}$. The difference of the ΔE_{micro} , almost keeps constant. However, this difference almost does not affect the linear relationship with a constant factor 130 between the surface friction and $(\Delta E_{\text{micro}})^2$ (see Fig R1), thus the conclusions is not affected.

Table R1. Fluctuations of E_{micro} dependent on different area with $S = 0.09 \text{ \AA}^2$, 0.16 \AA^2 , 0.25 \AA^2 , 0.36 \AA^2 , 0.49 \AA^2

	ΔE_{micro}				
	0.09 \AA^2	0.16 \AA^2	0.25 \AA^2	0.36 \AA^2	0.49 \AA^2
$q = 0.00 \text{ e}$	0.082	0.071	0.064	0.061	0.056
$q = 0.05 \text{ e}$	0.103	0.094	0.088	0.085	0.081
$q = 0.10 \text{ e}$	0.147	0.139	0.134	0.131	0.127
$q = 0.15 \text{ e}$	0.208	0.201	0.195	0.191	0.187
$q = 0.20 \text{ e}$	0.311	0.303	0.297	0.292	0.286
$q = 0.25 \text{ e}$	0.499	0.491	0.484	0.475	0.467
$q = 0.30 \text{ e}$	0.800	0.791	0.782	0.768	0.757

Fig. R1. Linear relationship between the surface friction coefficient and the fluctuation in microscopic potential energy, $(\Delta E_{\text{micro}})^2$, for the hexagonal lattice surfaces for various charges with several different areas $S = 0.09 \text{ \AA}^2$, 0.16 \AA^2 , 0.25 \AA^2 , 0.36 \AA^2 , and 0.49 \AA^2 , respectively.

4) What long-range contribution corrections were employed for dispersion and electrostatics "The authors now state that they used Ewald summation for long-range electrostatics; they do not comment on long-range dispersion. This should be clarified (i.e., stated in the manuscript). I have no objection to neglecting long-range dispersion in this case, but please say so if you do it, so that readers of the paper can understand what it is that you did here

Response : Thanks for the suggestions. We have used the cutoff for the dispersion when we treat the van der Waals interactions. The long-range dispersion interaction with a 1.0 nm cutoff was applied to the van der Waals interactions.

Change made: We have added this in our revised manuscript "A cutoff of 1.0 nm was used to calculate the dispersion (van der Waals) energies" in line 4 of the PS 6 "Simulation details and parameters we adopt" of our Supporting Information.

Reviewer #2 (Remarks to the Author):

This manuscript reveals an interesting fact that the surface frictions has little relationship with the surface wetting behaviors in the micro scale. The authors used molecular dynamics simulation to figure out the linear relationship of surface friction with square of energy fluctuation, which is also confirmed by the experimental atomic force microscopy method. I think this manuscript is suitable for publication in Communications Chemistry before the below concerns is addressed.

1. In line 43, the authors demonstrate the value less than 0.35e, but in line 49, they take an example of 0.36e. The authors should clarify this.

Response : Thanks for the suggestion. According to the suggestion, we simulated the system of the model charged surface (see Fig 1a in the revised manuscript) with the charge $q = 0.36 e$ and used the values of $0 e \leq q \leq 0.36 e$ in our model surfaces, where the $q = 0.36 e$ is equal to the charge of S atoms in MoS₂ surface. As shown in Figure R2, the friction is $4.8 \times 10^5 \text{ N s/m}^3$ for $q = 0.36 e$. The new results are shown in Fig. 1 in the revised manuscript.

Fig R2. Friction coefficient λ obtained from the correlation of friction forces

versus time t for $q = 0.36 e$.

2. The method of calculating E_{micro} should be included in the manuscript, at least introduced in supporting information.

Response: Thanks for the kind suggestions. We have introduced the method to calculate the E_{micro} as shown in the revised manuscript.

Change made: The new sentences reads “The large variation in friction coefficient is due to the large fluctuation of the localized potential energy, defined as $(\Delta E_{micro})^2 = \langle (E_{micro} - \langle E_{micro} \rangle)^2 \rangle$, which is obtained from the standard deviation of the microscopic interaction energy E_{micro} under inhomogeneous charge distributions.” in line 1, page 7 of the revised manuscript.

3. In the work of Xu et al, they constructed similar surface model, but test the nanofluidic behaviors. In that work, the surface charge ranges from 0.0 to 3.0e, but their surface friction, which is calculated from non-equilibrium MD simulations, does not change as much as this work demonstrates. The authors might give some discussions on this. (Journal of Physical Chemistry C, 2018, 122: 15772-15779.)

Response: Thanks for the comments. We have carefully read the paper by Xu et al. [*J. Phys. Chem. C*, 122, 15772 (2018)] and cited this paper in the revised manuscript. As the charge increases from 0 e to 2.0 e, the friction also increased from $(1.33 \pm 0.038) \times 10^4$ Ns/m³ to $(236.10 \pm 14.02) \times 10^4$ Ns/m³. This can be attributed to the short dipole length (length of the positive and negative charge) on the solid surface as Xu’s work, which decrease the energy fluctuations $(\Delta E_{micro})^2 = \langle (E_{micro} - \langle E_{micro} \rangle)^2 \rangle$. We have tested the solid surface similar as that in the paper [*J. Phys. Chem. C*, 122, 15772 (2018)] with charge $q = 0 e$ and 0.3 e. For comparison, we also present the results for our solid surface with charge $q = 0 e$ and 0.3 e (Fig R3a). Obviously, the surface friction for $q = 0.3 e$ with our solid surface is significant larger than that with the solid surface similar as Xu’s work. It originates from the larger fluctuation in the microscopic potential energy as shown in Figure R3b. These results can well explain the phenomenon that the larger dipole length of the solid surface would increase the

energy fluctuation, and then increase the surface friction remarkably.

Fig. R3. (a) Friction coefficient λ obtained from the correlation of friction forces versus time t and (b) the fluctuation in microscopic potential energy for the solid surface from our work and Xu's work with the charge $q = 0 \text{ e}$ and 0.3 e , respectively.

4. The measurement of contact angle in MD simulations are still controversial. Since the wetting behavior is somehow determined by the pressure (Research, 2019, 2019: 10.). Did authors consider the influence of pressure on the contact angle values? Or please convince me that the pressure takes no effect in this work.

Response: Thanks for the comments. We have carefully read the paper raised by the referee and cited this paper. The paper focused on the water molecules transporting through interlayers of two-dimensional nanosheets with various hydrophilicities using nonequilibrium molecular dynamics. They revealed that there is a threshold pressure drop $(\Delta P)_T$, exceeding which stable water permeability appears. They showed that there was a tight relationship between the surface hydrophobicity (measured by the contact angles) and the pressure difference between two boxes fully of water separated by the nanochannels, where water molecules are confined in-between.

We believe that this is different from the simulations in our work. Please kindly note that the pressure in our simulation does not affect the contact angles since the pressure in all our simulations to measure contact angles are constant. In our simulations, we used the NVT ensemble with constant volume and number and

constant temperature. The water droplet is at phase coexistence with the vapor at $T = 300$ K. The v-rescale method was used to control the temperature. The selection of the vapor-liquid coexistence system is to maintain the pressure in our simulation systems of the saturation vapor pressure [T. Koishi *et al.*, *Phys. Rev. Lett.*, 93, 185701 (2004)]. According to the tutorial by S. Babin [Water Vapor Myths: A Brief Tutorial], the vapor pressure only depends on the temperature, which is a constant controlled by the temperature coupling. Thus, all our simulations on the contact angle measurements keep the same pressure, that is the saturation vapor pressure at $T = 300$ K. As we know, this NVT setting is the mostly used in the MD simulation in measuring the contact angles [Rafiee, J., Mi, X., Gullapalli, H. *et al.*, Wetting transparency of graphene. *Nature Mater.* 11, 217 (2012) or Zhu *et al.*, *Proc. Natl. Acad. Sci. U. S. A.*, 113, 12946 (2016)].

5 I am interested in the change of surface frictions if the surface atom density is increased or decreased. Can authors give some discussions on this? The simulation data are not required to provide.

Response: Thanks for the suggestions. We have tested the dependence of the surface friction on the atom density. The lattice constant of the solid surface in our work was set to be 0.142 nm. We take the $q = 0.3$ e for example. Another two lattice constant (0.130 nm and 0.150 nm) are chosen to investigate the change of surface friction on the surface atom density. As shown in Figure R4a, when the lattice constant l increase from 0.130 nm to 0.150 nm, which means that the surface atom density is decreased, the friction increases from 2.3×10^5 Ns/m³ to 4.2×10^5 Ns/m³. Meanwhile, as shown in Figure R4b, the fluctuation in microscopic potential energy ΔE_{micro} increases from 0.449 kJ/mol for $l = 0.130$ nm to 0.782 kJ/mol for $l = 0.142$ nm and to 1.085 kJ/mol for $l = 0.150$ nm, which is well consistent with the trend for the change of surface friction. In a word, if the surface atom density is decreased, the surface friction would increase, and vice versa.

Fig. R4. (a) Friction coefficient λ obtained from the correlation of friction forces versus time t and (b) the fluctuation in microscopic potential energy for $l = 0.130$ nm, 0.142 nm, and 0.150 nm, respectively.

6. The acronym of HOPG should be introduced in the text of manuscript although it appears in Figure 1.

Response: Thanks for the suggestions. We have followed the suggestion and revised the manuscript.

Change made: We have revised the manuscript in the main text, which reads “*highly oriented pyrolytic graphite (HOPG)*” in line 1 page 9 in the revised manuscript.

Reviewer #3 (Remarks to the Author):

This is an interesting paper discussing the effect of surface charging on friction. The obtained results are useful, but the information is not surprising and particularly innovative. On the other hand, the obtained correlations between various system parameters are carefully examined, which makes this paper a nice reference work. Overall, I feel that the paper would fit better to Scientific Reports.

Response: We thank the referee for the positive comments on our work that “This is an interesting paper discussing the effect of surface charging on friction”.

We note, generally, the microscopic friction is determined by measuring the

contact angle; a large contact angle indicating a hydrophobic surface is associated with low surface friction, and vice versa. However, in this paper, using molecular dynamics simulations, we unexpectedly find even over an order of magnitude difference of the friction coefficient at the small charge difference ($q \leq 0.36$ e) on the two-dimensional material and biological lipid surfaces, despite of the similar contact angle values on these surfaces as traditionally thought. This large difference is confirmed by experimentally measuring surface friction of graphite and MoS₂ using atomic force microscopy (AFM). The large variation of friction coefficient is attributed to the significant fluctuations of localized potential energy profile in presence of the inhomogeneous charge distributions.

Interestingly, we note that the most of popular two-dimensional materials and biomolecules (67.1% in all the residues atoms listed in OPLSAA force field) are formed by the atoms or groups with the small charges ($q \leq 0.36$ e). For example, the carbon atoms in nitrogen-doped and the hydrogen terminated graphene usually attain the charge less than 0.2 e, the carbon atoms of terminal methyl of the lipid usually attain the charge of -0.18 e and the carbon atoms of benzene rings in Phe of proteins residues can attain the charge of -0.115 e, and the S atoms of MoS₂ attain the charge of 0.36 e. While, these small charges atoms/groups on the material and biomolecular surfaces are usually neglected in understanding the surface dynamics properties. This may be attributed to the negligible electrostatic interaction energy contribution of the total surface/water interactions due to the small charges.

Our results herein reveal that this general knowledge of wetting behaviors are not directly applied to infer the dynamics behaviors in respect of the frictions on the surfaces of the most popular two-dimensional materials and biomolecules. Our findings not only open a door to design frictionless nano-devices, but also represent a big step towards understanding the microscopic nature of the dynamics binding processes in protein-ligand in the crowded cell environments, molecular through the lipid membranes and the protein folding rates related to the internal frictions.

In a word, we think that this paper will make important contributions and the achieved results should be of broad interest to a large community of scientists in the

fields of chemistry, materials, physics, biology science and nanotechnology, such as the theory of the molecular frictions, folding of the protein molecules, fabricating/manipulating of nano-materials. We believe that Communications Chemistry is the right journal for its publication.

REVIEWERS' COMMENTS:

Reviewer #1 (Remarks to the Author):

The authors responded to all comments correctly and I think the revised manuscript has improved than the previous one. As I mentioned in the early version, this work is interesting and I recommend it for publication.

Reviewer #2 (Remarks to the Author):

The authors addressed all my concerns in their revised manuscript. I recommend the acceptance of this manuscript in Communications Chemistry.